# *Ceratocystiopsis quercina* sp. nov. Associated with *Platypus cylindrus* on Declining *Quercus suber* in Portugal

**DOI:** 10.3390/biology11050750

**Published:** 2022-05-13

**Authors:** Maria L. Inácio, José Marcelino, Arlindo Lima, Edmundo Sousa, Filomena Nóbrega

**Affiliations:** 1Instituto Nacional de Investigação Agrária e Veterinária, Quinta do Marquês, 2780-159 Oeiras, Portugal; edmundo.sousa@iniav.pt (E.S.); filomena.nobrega@iniav.pt (F.N.); 2GREEN-IT Bioresources for Sustainability, Instituto de Tecnologia Química e Biológica, Universidade Nova de Lisboa (ITQN NOVA), Av. da República, 2780-157 Oeiras, Portugal; 3Entomology and Nematology Department, University of Florida, Gainesville, FL 32608, USA; jmarcelino@ufl.edu; 4LEAF—Linking Environment Agriculture and Food, Instituto Superior de Agronomia (ISA), University of Lisbon, Tapada de Ajuda, 1349-017 Lisbon, Portugal; arlindolima@isa.ulisboa.pt; 5LPVVA—Laboratório de Patologia Vegetal “Veríssimo de Almeida”, Instituto Superior de Agronomia (ISA), University of Lisbon, Tapada da Ajuda, 1349-017 Lisbon, Portugal

**Keywords:** *Ophiostoma*, ambrosia fungi, ambrosia insect, cork oak, *Quercus suber*, Portugal

## Abstract

**Simple Summary:**

The presence of the oak pinhole borer, the insect *Platypus cylindrus*, in Portuguese cork oak stands has drastically increased in the past few decades. This beetle excavates long galleries in the trunk while inoculating fungi (called ambrosia fungi) transported in special organs (mycangia) that will serve as food source for its offspring. The combined action of extensive boring into the heartwood and the inoculation of fungi leads to an increase in tree mortality. A new ambrosial fungus, *Ceratocystiopsis quercina*, was isolated from the insects’ mycangia and from wilting trees, namely from the staining patches it causes on wood.

**Abstract:**

*Platypus cylindrus* is the most common ambrosia beetle in stands of *Quercus suber* in Portugal. This insect farms specialized fungi in sapwood galleries, using its mycangia to carry and store these organisms. Some ectosymbiotic fungi carried by *P. cylindrus* are phytopathogenic and cause extensive tree mortality and severe economic losses. To understand the role of *P. cylindrus* fungal symbionts in stands of *Q. suber* we examined beetle galleries present in declining and/or dying cork oak trees during field surveys. Logs with active galleries were obtained in situ and from captured emerging beetles. Insects were aseptically dissected, and their mycangia and intestine were retrieved. Morphological and molecular profiles of fungal isolates obtained from cultured insect parts were carried out to accurately characterize and identify isolated fungi. Molecular characterizations were performed with DNA sequence data from four loci, i.e., LSU, SSU, 5.8S-ITS2-28S, and TUB. Morphological results consistently showed a collection of *Ophiostoma*-like fungal axenic isolates, while phylogenies inferred that this collection constitutes an undescribed taxon reported herein for the first time in association with *P. cylindrus* in Portuguese cork oak stands. The novel species was erected as *Ceratocystiopsis quercina* sp. nov. and constitutes a new phytopathogenic fungal species associated with symptoms of vegetative cork oak decline.

## 1. Introduction

*Platypus cylindrus* Fab. (Coleoptera: Curculionidae) is a ubiquitous ambrosia beetle in stands of *Quercus suber* L. in the Mediterranean basin. This insect was initially considered a secondary pest in Portuguese stands as its impact was generally limited to dead or weakened trees [1,2,3]. However, since the 1980s, severe infestations were observed in apparently healthy cork oaks, causing widespread tree death within three months to one year and a half after infestation [4,5,6]. Cork oak forests are a very specific, delicately balanced ecosystem which mainly persists in the Mediterranean basin. It is therefore of major concern that over the last three decades an alarming decline of trees has increased across its distribution area, namely in the representative Portuguese cork oak stands. *Platypus cylindrus* emerged as a determinant factor in the decline of stands since adults attack trees of all ages, especially those recently decorked or weakened, but retaining wood humidity. The attacks of *P. cylindrus* are localized in the trunk and branches of larger diameter (Figure 1a,b). The males begin the colonization of trees through gallery excavations, while the females carry fungi, called ambrosia fungi, in specialized structures in their thorax, known as mycangia [7,8,9,10] (Figure 1c). These symbiotic fungi will grow on the walls of the galleries providing not only a significant food source to both insect larvae and adults, but also giving the fungi a continuous mean of dispersal [11] (Figure 1d,e). *Platypus cylindrus* lives in close association with ambrosia fungi, which in turn are a determinant factor in the decline of cork oak stands, causing severe economic losses in the Mediterranean region [12,13]. The most important group of ambrosia fungi are members of the heterogeneous Ophiostomatales group, a so-called ophiostomatoid fungi. These include genera with similar morphological and ecological characteristics, i.e., *Ceratocystiopsis*, *Graphilbum*, *Leptographium*, *Ophiostoma*, *Raffaelea,* and *Sporothrix* [14,15]. The taxonomy of ophiostomatoid fungi was intricate since the first descriptions were reported [16,17,18]. At present they are grouped in two fungal families, i.e., Ceratocystidaceae and Ophiostomataceae. The genus *Ceratocystiopsis* includes nearly 20 taxa, most of which obtained from plants infested by phloem-and-wood-breeding beetles.

Several ophiostomatoid fungi have already been reported to be associated with *P*. *cylindrus* and its galleries, in *Quercus* spp. in France, Algeria, Greece, Portugal, and Tunisia, namely the *Raffaelea* species [19,20,21,22,23,24,25,26,27]. Studies of oak decline in Europe have also shown that the fungi complex *Ophiostoma/Ceratocystis* was pathogenic to *Quercus* trees [28,29,30]. In this study, the newly collected *Ophiostoma*-like fungi isolates were accurately identified based on morphological characters and DNA sequences for four loci (LSU, SSU, 5.8S-ITS2-28S and TUB) [31,32,33,34,35,36,37,38,39,40,41,42,43,44,45] with the overall objective of contributing to the knowledge of the etiology of cork oak decline in Portuguese stands.

## 2. Materials and Methods

### 2.1. Sampling and Fungal Isolation

In total, 12 cork oaks infested by *P. cylindrus* and exhibiting vegetative decline symptoms were selected in 2 main cork-producing regions of Portugal, i.e., Ribatejo province and Alentejo province. One log from each tree (30 cm diam. and 0.5 m length) was collected and the sampling was repeated during 2005, 2006, and 2007. Beetles were captured from active galleries with fine mesh nets attached to the logs and observed under a binocular microscope to confirm their identity. A total of 100 insects per year were aseptically dissected with sterilized ophthalmic scalpels (Feather ^®^, Sacramento, CA, USA) under a stereo binocular microscope × 40 LEICA MZ6 (Wetzlar, Germany) to obtain their mycangia, intestine, and parts of the exoskeleton (elytra).

Collection of beetles was followed by sampling of fungi from gallery walls by cutting logs in sections exposing the gallery system where *P. cylindrus* larvae feed. Wood fragments containing galleries were excised from the logs. Fungal isolation from the beetle parts and wood fragments was made after surface sterilization of these pieces for 1 min in 1% sodium hypochlorite. Surface sterilized samples were subsequently plated directly on 1.5% malt extract agar (10 g Difco MEA, Franklin, NJ, USA; 15 g agar in 1 L dH_2_O) amended with 500 mg/L cycloheximide (Sigma-Aldrich, St. Louis, MO, USA) and 500 mg/L streptomycin (Sigma-Aldrich, St. Louis, MO, USA). Plates were incubated at room temperature in dark conditions for 1 to 2 weeks, when visible fungal growth was observed and axenic cultures of each putative strain were obtained.

### 2.2. Microscopic Observation and Descriptions

Colonized agar plugs (5 mm diam.) were excised from actively growing 1 week old pure cultures of different isolates. These discs were transferred to the center of fresh plates containing 1.5% MEA. Growth rates were determined at temperatures ranging from 5 to 35 °C, at 5-degree intervals, 3 and 10 days after inoculation, in the dark. The colony diameter of six replicates was calculated by averaging the 12 measurements. Mycelial colours were described using the terminology from Saccardo (1891) [46]. Tolerance to cycloheximide was assessed by measuring fungal growth on MEA amended with 100, 500, and 1000 ppm cycloheximide after autoclaving. For fungal morphological characterization, 3 to 5-day-old slide cultures mounted in lactophenol were examined with light microscopy with differential interference contrast microscopy (Olympus BX-41 with Olympus DP11, Tokyo, Japan) [47]. Fifty measurements were obtained for each taxonomically informative structure. For scanning electron microscopy (SEM), small wood blocks (5 × 2 × 5 mm) bearing fungal structures were fixed according to Lee et al., (2003) [48] and Massoumi-Alamouti et al. (2009) [44]. After fixation, the samples were critical point dried, sputter coated twice with gold palladium (98:2), and examined using a JEOL 35 scanning electron microscope (JEOL, Peabody, MA, USA). Isolates used in this study are maintained at the Centraalbureau voor Schimmelcultures (CBS), Utrecht, The Netherlands, as well as in the culture collection of INIAV Institute (Micoteca da Estação Agronómica Nacional (MEAN)) (PC acronyms, Table 1). In addition, mycelial plugs were placed in the voucher specimen collection at Iowa State University (secondary collection; C acronyms, Table 1). Voucher information and GenBank accession numbers of all isolates included in this study and sequences used in the phylogenetic analyses are listed in Table 1.

Based on unique culture morphology, representative isolates were selected for DNA sequence-based characterization. Fungal DNA extraction was performed using mycelia from pure cultures with the Puregene^®^ DNA Purification Kit (Gentra Systems Inc., Minneapolis, MN, USA) according to the manufacturer’s instructions. Five genomic regions were amplified by polymerase chain reaction (PCR) and sequenced for phylogenetic analyses. The nuclear large subunit ribosomal DNA (LSU, 28S rDNA) was amplified using primers NL1 and NL4 [49] as well as LROR and LR5 [49,50]. The nuclear small subunit ribosomal DNA (SSU, 18S rDNA) was amplified with primers NS1, NS3, NS4, and NS6 [51,52]. The internal transcribed spacer 1 and 2 (ITS1-5.8S-ITS2) and the internal transcribed spacer 2 and large subunit (5.8S-ITS2-28S) were amplified with primers ITS5/NL4 [52], ITS1F [53], and ITSp3 [54]. Amplification of β-Tubulin (TUB) used primers T10 [55] and Bt2b [56]. All reactions run in a 25 μL volume, containing 12.5 μL of Supreme NZYTaq II DNA polymerase Master Mix (NZYTech,, Lisbon, Portugal), 2 μL of DNA template, 1 μL of each forward and reverse primer (10 μM) and 8.5 μL of molecular-grade water (Sigma-Aldrich, St. Louis, MI, USA). Amplification reactions were performed in the thermocycler Biometra TAdvanced (Analytik, Jena, Germany). Amplification of the various loci was performed under the following conditions: a denaturation step at 95 °C for 5 min followed by 35 cycles at 94 °C for 1 min, 1 min at 50–55 °C (depending on the primer annealing temperature), and 1 min at 72 °C, with a final extension step of 7 min at 72 °C. Amplified products were visualized under UV light on a 1.5% agarose gel to confirm successful amplification. PCR products were purified using ExoSAP-IT™ PCR Product Cleanup Reagent (ThermoFisher Scientific, Pittsburg, PA, USA) following the manufacturer’s protocols. These were submitted to the Sequencing facility at STABVIDA (Caparica, Portugal) for Sanger sequencing. Consensus sequences were assembled using Sequencher^TM^ (Gene Codes Corp., Ann Arbor, MI, USA). Consensus sequences were trimmed and a preliminary molecular identification was made by comparing the sequences of our isolates with those of the National Center for Biotechnology Information (NCBI) using the Basic Local Alignment Search Tool (BLAST: https://www.ncbi.nlm.nih.gov, accessed on 5 May 2021) and those in datasets referred by other authors [17,44,49,50,57]. Phylogenetic analyses were conducted separately for the five rDNA regions (nSSU, nLSU, ITS1-5.8S-ITS2, 5.8S-ITS2-28S, and TUB) with new sequences generated in this study and with other selected published sequences based on their genetic distance to Ophiostomatales [58,59,60]. Novel sequences from this study were deposited in GenBank (accession numbers are included in Table 1). A concatenated dataset analysis of the three adjacent regions was not done due to the high degree of divergence of sequences from our isolates and other Ophiostomatales available in GenBank (variance and length of the ITS1 region). We opt for a discrete phylogenetic analysis of the 5.8S-ITS2-28S region and partial nucleotide sequences from the large and small subunit rDNA, LSU, and SSU, and from β-tubulin, TUB. Only sequence fragments that could be aligned with certainty were used to generate alignments and included in the maximum likelihood phylogenetic analyses with MEGA, version X [61] with 1000 bootstrap replicates. The best evolutionary substitution model for LSU and ITS regions was Tamura 3-parameter (T92+G) and for SSU and TUB regions was Kimura 2-parameter (K2+G). Only bootstrap values above 50 were considered well supported in the final consensus tree.

## 3. Results

### 3.1. Fungal Isolation and Identification

Fungal isolations were performed from 300 adults *P. cylindrus* emerged from the cork oak logs and from pieces of their galleries during sampling seasons of three years. The most frequent fungi were species of the ophiostomatoid Raffaelea genus, including a novel species erected as *Raffaelea quercina* Inácio, Sousa, and Nóbrega (2021), a new pathogenic fungus associated with *P. cylindrus* [62]. Other *Ophiostoma*-like colonies were the second most frequent species, being mainly present in the mycangia in the insect and also prevalent on the intestine and on the ambrosial mat lining of the galleries. A total of seven axenic *Ophiostoma*-like colonies were selected for further molecular analyses and morphological characterization.

### 3.2. Phylogenetic Analyses

Seven unidentified *Ophiostoma*-like isolates were phylogenetically placed using four loci (LSU, SSU, 5.8-ITS2-28S, and TUB) corresponding to four nuclear genes, i.e., 28S rDNA, 18S rDNA, ITS2, and β-Tubulin. We aimed to identify isolates at the species level.

Datasets for phylogenetic analyses included available sequences for reference species in the genus *Ceratocystiopsis,* as well as other representative taxa in the *Ophiostoma* genus (Table 1). The first aligned dataset (Figure 2a), contained LSU 28S rDNA sequences of *Ceratocystiopsis* species and related species in closely related genera (451 characters, 82 parsimony informative). The second alignment (Figure 2b), contained SSU 18S rDNA sequences (1024 characters, 51 parsimony informative). For the phylogenetic analysis of the 5.8S-ITS2-28S region (Figure 2c), the alignment had 311 characters, including gaps, of which 116 were parsimony informative. The fourth alignment (Figure 2d), contained the β-tubulin (TUB) sequences (242 characters, 133 parsimony informative).

The seven isolates from this study, included in all the separate analyses of the rDNA and β-Tubulin gene regions, resulted in trees with similar topologies (Figure 2). The novel fungal isolates derived from this study claded apart from all pre-existing and known clades (57–89% bootstrap support. See Figure 2).

The phylogenic analyses presented in this study support the existence of a new *Ophiostoma*-like species clearly discriminated from pre-existing species and clades. The different phylogenetic trees consistently group the Portuguese isolates in a single and distinct clade, closely related to *Ceratocystiopsis* species, therefore supporting the novel status of these isolates as a distinct new species of *Ceratocystiopsis*, herein described and referred to as *Ceratocystiopsis*
*quercina* sp. nov.

### 3.3. Morphology and Taxonomy

*Ceratocystiopsis quercina* M.L. Inácio, E. Sousa, and F. Nóbrega, sp. nov (Figure 2).

(1)MycoBank: MB 841995;(2)Holotype: LISE 96335;(3)Etymology: Named after the host genus from which it was isolated, *Quercus;*(4)Host trees/distribution: On galleries of *Quercus suber* in Portugal = on mycangia of *Platypus cylindrus.*

Material examined: Portugal, Chamusca (Santarém), on mycangia and in galleries of the insect *Platypus cylindrus* on declining *Quercus suber*, Maria L. Inácio, May 2006 (LISE 96335 holotype; ex-type culture PC05.032 = MEAN 1336 = C2508 = CBS 148604); Portugal, Chamusca (Santarém) in galleries of the insect *Platypus cylindrus* on declining *Quercus suber,* Maria L. Inácio, May 2005 (living culture, PC05.005 = MEAN 1335 = C2510 = CBS 148603); Portugal, Chamusca (Santarém) in the mycangia of *Platypus cylindrus* emerged from *Quercus suber*, Maria L. Inácio, May 2005 (living culture, PC05.006 = MEAN 1297 = C2511); Portugal, Chamusca (Santarém) in the mycangia of *Platypus cylindrus* emerged from *Quercus suber*, Maria L. Inácio, May 2006 (living culture, PC06.022 = MEAN 1298 = C2519); Portugal, Montemor (Alentejo) in the mycangia of *Platypus cylindrus* emerged from *Quercus suber*, Maria L. Inácio, May 2006 (living culture, PC06.034 = MEAN 1337 = C2507); Portugal, Montemor (Alentejo) in galleries of the insect *Platypus cylindrus* on declining *Quercus suber,* Maria L. Inácio, May 2007 (living culture, PC07.004 = MEAN 1338 = C2517); Portugal, Montemor (Alentejo) in the intestine of the insect *Platypus cylindrus* emerged from *Quercus suber,* Maria L. Inácio, May 2007 (living culture, PC07.007 = MEAN 1299 = C2509 = CBS 148605).

Description: Colonies effuse, yeast-like, ivory-white to cream-colored, smooth, later mucilaginous, with light concentric zonation, few with a light olive-green mottling appearing in the center or a sporodochium-like both in culture and in wood (Figure 3a–d), corresponding to a *Hyalorhinocladiella* anamorph which initially formed protoperithecia in culture that did not developed necks even when isolates were paired (Figure 3e,f); perithecia develop abundantly in 30 days, dark brown, superficial on wood and in culture on the superficial mycelium; globe-shaped, (55-)60-70(-80) μm diameter; neck short (15–25 μm long), conical, with an obtuse apex. Ascospores extrude through the ostiole in a narrow cirrhus; hyaline, one-celled, fusiform with a hyaline sheath, (9.2-)10.0-11.3(-12.2) × (0.8-)1.0-1.2(-1.4) μm (Figure 3H–K). Colonies grow slowly on MEA, 37–38 mm after 10 d at 25 °C. Hyphae hyaline and septate that bound together forming compact hyphal ropes with cluster of conidia. Conidiophores micronematous and mononematous or synnematous, erect, septate, slender with a tapered apex. Conidial development occurring through both annellidic percurrent or sympodial proliferation but not leaving conspicuous scars (Figure 3G,L–O). Conidia with various shape being the triangular the most prevalent, with (4.2-)5.2-5.8(-8.4) × (1.7-)1.8-2.2(-3.3) μm (Figure 3P–Q). Our SEM micrographs do not help in the clarification on the mode of conidial development since we found hyaline conidiophores and primary annellidic conidiogenous cells as well as proliferation sympodial.

## 4. Discussion

This study identified, characterized, and erected a new species of ambrosia fungus with a *Hyalorhinocladiella* anamorph associated with *Platypus cylindrus* in declining cork oak trees in Portugal. Morphological features and phylogenetic analyses supported the assignation of the axenic isolates retrieved during this study to a novel species. The species was named *Ceratocystiopsis quercina* based on clade relationships with other *Ceratocystiopsis* species.

*Ceratocystiopsis quercina* was found closely associated with the mycangia of *P. cylindrus,* being also isolated from their intestinal tract. Genome fragment sequences of the cycloheximide-tolerant *Hyalorhinocladiella* isolates did not match known available genome sequences. These divergences corroborated the differences found in the morphology, i.e., the absence of sprout cells or sporodochia typical of *Raffaelea* species, common symbionts of *P. cylindrus*. Based on these findings, the species is new to science, and is the first known *Ceratocystiopsis* associated with this ambrosia beetle.

To date, only *Raffaelea* spp. and *Ambrosiella* spp. have been documented as alimentary mycangial symbionts of ambrosia beetles. The sexual state of some of these ambrosia beetle symbionts was only found recently [63,64]. Previously, it was believed that a sexual state would not be an advantage for dispersal by an ambrosia beetle with mycangia [50]. It is probable that species of *Raffaelea* have derived from an ancestor with an ophiostomatoid sexual state and conidiogenesis similar to extant species of *Hyalorhinocladiella* or *Pesotum* [60]. The accurate isolation technique from the mycangia and intestinal content of the beetles employed in this study, and the use of cycloheximide in the isolation medium allowed a good recovery of the ophiostomatoid symbionts. *Ceratocystiopsis quercina* found in association with *P. cylindrus* was also isolated from the galleries of the insect in declining oaks. It was also recovered from declining cork oaks with visible aborted attacks of *P. cylindrus*. Thus, even if insects do not succeed in colonizing the tree, they are able to inoculate the pathogen into a susceptible host. Without *P. cylindrus* as a vector for dissemination, it would not be possible for these fungi to reach new hosts, as they are enclosed within the tree. In addition, fungi in the Ophiostomatales require pre-existing wounds in order to infect their hosts. The beetles enable infection by carrying the fungi into pre-existing wounds on trees or produce these wounds themselves while excavating galleries [65]. Thus, without *P. cylindrus*, it would not be possible for the fungal species to continue its life cycle, just as without the fungi, the beetles would have an extremely difficult time colonizing new trees. It is, indeed, an obligatory symbiosis. Inside the host, the fungus can rapidly spread from several points.

In the wood of their host trees, ambrosia fungi usually penetrate only a few mm into the xylem and their growth is usually restricted to areas surrounding the galleries [66]. However, *C. quercina* penetrates several cm into the sapwood of its cork oak hosts and causes a brown discoloration in the xylem. It is likely that, as it occurs with fungi such as *R. lauricola*, aborted attacks by the insect in the sapwood of healthy trees vector thousands of spores of the fungus, which oozees from the mycangia of the beetles, infecting severed vessels, and ultimately cause the systemic colonization of the host [67,68]. This most likely new strategy allows the insect to spread wilt disease in cork oak stands, facilitating the beetles’ establishment in the host plant. In this manner, *P. cylindrus* does not need to wait another decorking cycle to establish new populations in *Q. suber* stands, and trees could be potentially colonized from the decorking until they rebuilt a thick cork layer. In addition, the beetle’s success has most likely been further enhanced with climate change, with continuous mild winters causing less offspring mortality, and summer droughts causing stressed trees, thus making host trees more susceptible to attacks [69].

It has generally been accepted that one or a few fungal species are associated with a particular ambrosia beetle species, however, more recent studies note that fungal symbionts of ambrosia beetles are more diverse, more generalist and more competitive than previously assumed, and that ambrosia fungi may compete among each other for entrance to, and growth within, the mycangia of the vectoring beetles [50,63]. If so, it would be expected that more species could be isolated from *P. cylindrus*, especially if collected in other parts of its distribution range. Additionaly, new taxa of Ophiostomatales are being revealed and more comprehensive and robust phylogenies are being provided [70].

In terms of role in oak decline, the combined action of *P. cylindrus* massive attacks and extensive gallery excavation with a successful parallel inoculation of ambrosia fungi into the plant hosts, leads to an increase of tree mortality enhanced through new associations with wilt-causing fungi. Understanding the ecology and population dynamics of *P. cylindrus*-associated fungi is important for the surveillance and management of the beetle-fungal complex and it impacts on forest stands, and could improve prediction and modeling of disease dissemination. Biological control of these fungal phytopathogens may be possible through manipulation of the mycangial mycoflora. However, with the consistent isolation of Ophiostomatales. from all beetles sampled, the incidence of cork oak wilt appears to be driven only by the population level of *P. cylindrus*, and disease management should focus on this parameter rather than mycoflora manipulation.

## 5. Conclusions

Cork oak forests are very specific, delicately-balanced ecosystems that only persist in the Mediterranean basin. It is therefore of major concern that over the last three decades an alarming decline of trees has increased across its distribution area, namely in the representative Portuguese cork oak stands. Due to cork oak decline being a multifactorial process, several causes have been pointed out as contributors to tree mortality and loss of vigour, namely biotic factors. The insect *Platypus cylindrus* emerged as a determinant factor in the decline of stands and its population outbreaks in the last decades have caused heavy economic damages since cork loses its quality and ultimately trees death overcomes.

The symptoms and signs exhibited by cork oaks attacked by *P. cylindrus*, including the presence of numerous entry holes and profuse sawdust emerging from these holes, do not reveal the real dimension of the attack intensity within the trunk. Coupled with this extensive boring activity, the inoculation of ambrosia fungi (Ophiostomatales) is part of the insect’s strategy to establish its offspring in the host trees. Ambrosia beetles and their associated fungi constitute a small part of a much larger food web, the complexities of which we have barely started to understand. There are many questions about the extraordinary complexity of the interactions between these wood-inhabiting beetles, the assembly of fungi which they transmit, and the tree which supports the whole community. We believe that the research described herein with the discovery of a new ambrosia fungi, *Ceratocystiopsis quercina*, improves the knowledge on the mycobiota associated with the oak pinhole borer in Portuguese cork oak stands, and can help to avoid costly mistakes in the management of these emblematic stands, preserving the economic and cultural heritage of the unique cork oak stands and landscapes present in the Mediterranean.

## Figures and Tables

**Figure 1 biology-11-00750-f001:**
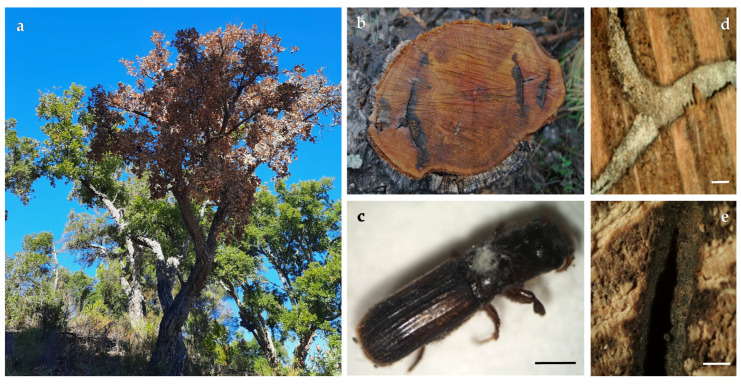
(**a**) Cork oak affected by *Platypus cylindrus* and its ambrosia fungi; (**b**) cross section of the trunk with wood staining along the galleries; (**c**) emerged beetle with the ambrosial mycelium coming out of the mycangia; (**d**) transversal cut showing the gallery lined by a silky white mycelium, (**e**) older gallery (bars = 1 mm).

**Figure 2 biology-11-00750-f002:**
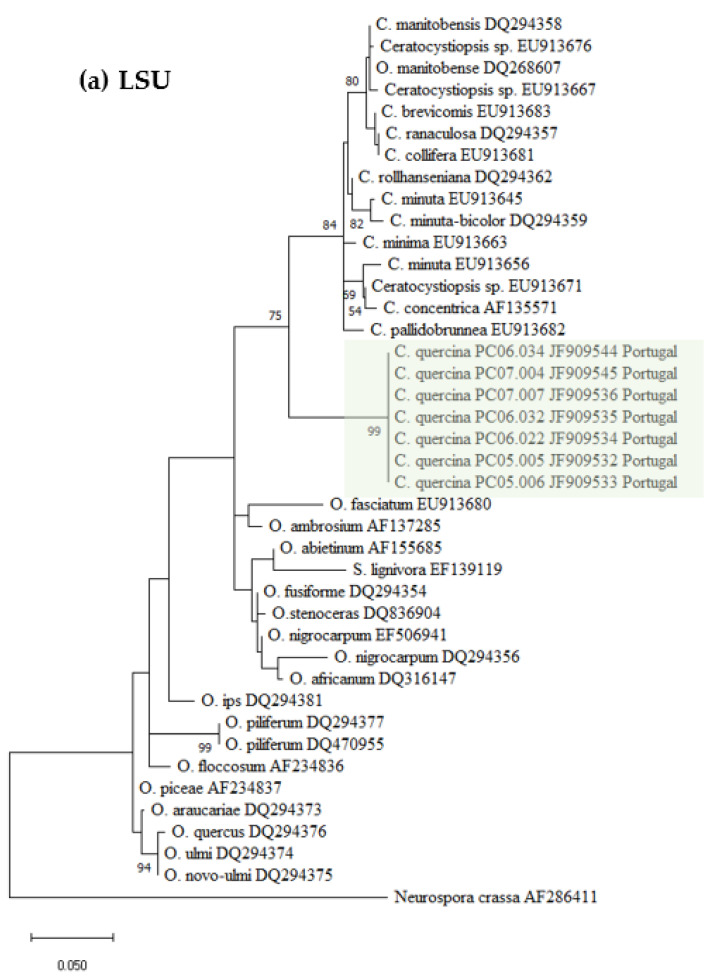
Phylogenetic trees resulting from maximum likelihood analyses of the (**a**) LSU, (**b**) SSU, (**c**) ITS2, and (**d**) TUB regions for species of *Ophiostoma* and *Ceratocystiopsis.* Bootstrap support values above 50 are indicated on the nodes. New sequences and new species proposed in this study are indicated in color. In all the phylogenies three clades were distinctly present, i.e., one represented by known isolates of *Ophiostoma* spp., a second one containing isolates of known *Ceratocystiopsis* spp., and a third clade with the new seven isolates from this study.

**Figure 3 biology-11-00750-f003:**
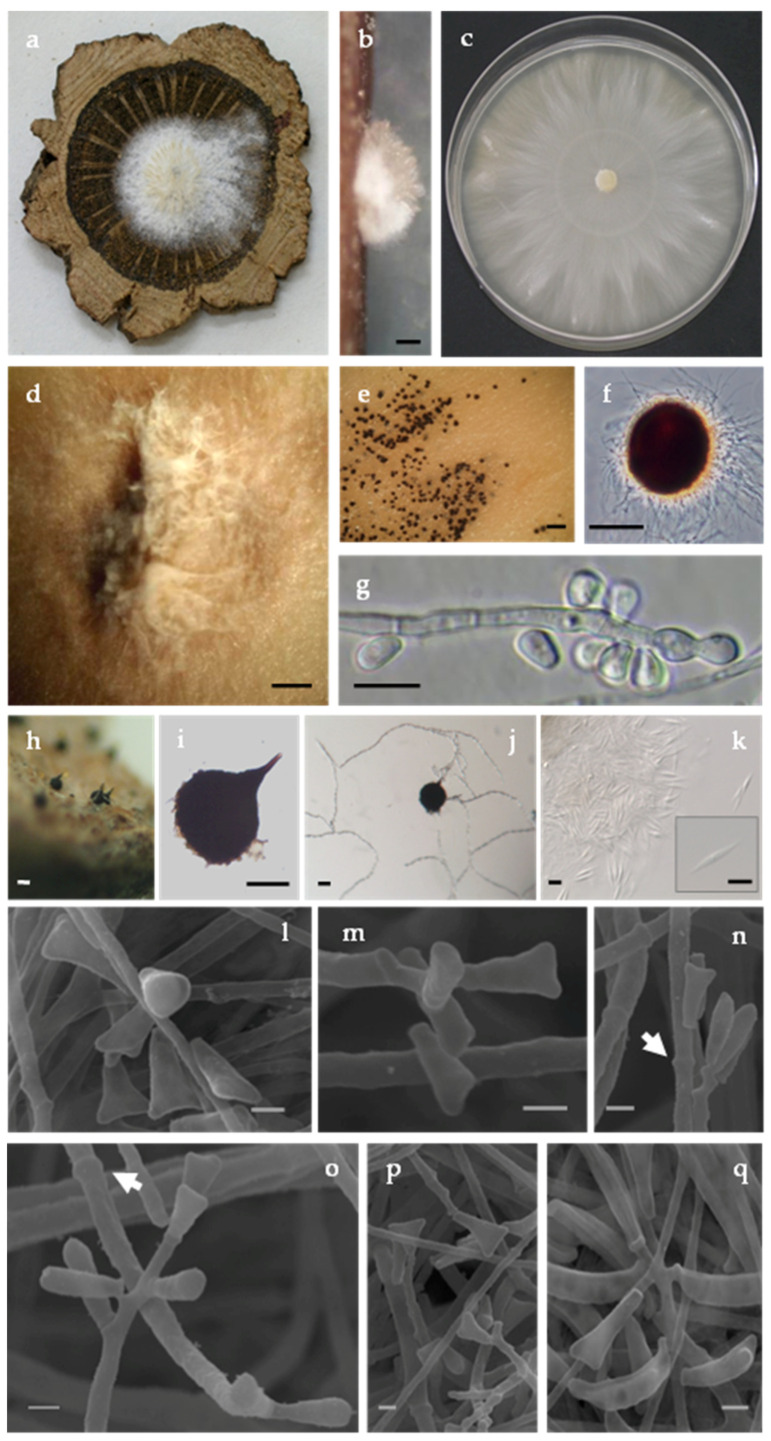
Culture, conidiophores, and conidia of *Ceratocystiopsis quercina* (isolate PC06.032 = C2508). (**a**) Culture growing on a wood disc of *Quercus suber*. (**b**) Sporodochia-like of conidiophores on wood (bar = 250 μm). (**c**) Colony morphology after 2 weeks on malt extract agar in 90 mm diameter plate. (**d**) Mass of conidiophores, conidia, and yeast-like growth on malt extract agar (**d**,**e** bar = 500 μm). (**e**,**f**) Protoperithecia formed on malt extract agar, (**f**) Differential interference contrast (bar = 50 μm). (**g**) Newly formed conidia through percurrent proliferation without conspicuous scars (annellations) at the point of conidial dehiscence (bar = 10 μm). (**h**–**j**) Perithecia formation in 30 days, (**h**) Superficial on autoclaved wood (bar = 50μm) and (**i**) on culture, (**j**) Ascospores extruding from the ostiole (**i**,**j** bar = 25 μm). (**k**) Ascospores fusiform with sheath (bar = 5 μm). (**l**–**q**) Scanning electron micrographs of conidia of various shape and conidiogenous with percurrent and sympodial proliferation and some conidiogenous cells showing annellations (arrows in (**n**) and (**o**)) (bar = 1 μm). Note: Morphological comparisons and growth data indicates that this *Ceratocystiopsis* species, with *Hyalorhinocladiella* anamorph, retrieved from *P. cylindrus* and their galleries in *Q. suber* is different from any Ophiostomatales previously described. This fungus could also be distinguished from previously described species based on DNA comparisons, as documented in this study. The new Portuguese *Ceratocystiopsis* species presents a stable and distinct position in the inferred phylogenetic relationships.

**Table 1 biology-11-00750-t001:** Details of isolates obtained in this study (bold) and of strains representing species of *Ophiostoma* and *Ceratocystiopsis* retrieved from GenBank and used in phylogenetic analyses.

Species	Source ^a^	Country	Associated Insect or Host	GenBank Accession nrs. ^b^			
				nSSU rDNA	nLSU rDNA	5.8S-ITS2	TUB
** *C. quercina sp. nov.* **	**PC05.005, C2510**	**Portugal**	** *Quercus suber* **	**JF909502**	**JF909532**	**JF909518**	**MZ450136**
** *C. quercina sp. nov.* **	**PC05.006, C2511**	**Portugal**	** *Platypus cylindrus* **	**JF909503**	**JF909533**	**JF909519**	**MZ450137**
** *C. quercina sp. nov.* **	**PC06.022, C2519**	**Portugal**	** *P. cylindrus* **	**JF909504**	**JF909534**	**JF909520**	**MZ450138**
** *C. quercina sp. nov.* **	**PC06.032, C2508**	**Portugal**	** *P. cylindrus* **	**JF909505**	**JF909535**	**JF909521**	**MZ450139**
** *C. quercina sp. nov.* **	**PC06.034, C2507**	**Portugal**	** *P. cylindrus* **	**JF909506**	**JF909544**	**JF909522**	**MZ450140**
** *C. quercina sp. nov.* **	**PC07.004, C2517**	**Portugal**	** *Q. suber* **	**JF909507**	**JF909545**	**JF909523**	**MZ450141**
** *C. quercina sp. nov.* **	**PC07.007, C2509**	**Portugal**	** *P. cylindrus* **	**JF909508**	**JF909536**	**JF909524**	**MZ450142**
*O. abietinum*	CBS125.89	Mexico	*Abies vejari*		AF155685	AY924382	
*O. abietinum*	CMW1468	Canada	*Dendroctonus ponderosa*				AY280468
*O. abietinum*	CMW109	USA	*Pinus echinata*				AY280469
*O. africanum*	CMW1104	South Africa	*Protea caffra*		DQ316147		DQ316162
*O. ambrosium*	CBS 210.64				AF137285		
*O. araucariae*	CMW671	Chile	*Araucaria araucaria*		DQ294373		
*O. araucariae*	CBS 114.68	Chile	*Araucaria araucaria*				KU184289
*O. arborea*	TUB F4270	Germany	*Ips typographus*	AY497511			
*O. aurorae*	CMW19364	South Africa	*Pinus elliottii* *Hylurgus angustatus*			DQ396798	DQ396802
*O. abieticola*	YCC478	Japan	*Ips subelongatus*			GU134156	
*O. bicolor*	TUB F4269	Germany	*Ips typographus*	AY497512			
*O. fasciatum*	UM56	Brit. Columbia	*Pseudotsuga menziesii*		EU913680		EU913759
*O. floccosum*	AU55-6				AF234836		
*O. floccosum*	CNB 117A	Spain	*Pinus pinaster*				
*O. floccosum*	CMW 171	USA	*Pinus ponderosa*				DQ296087
*O. floccosum*	387N	Canada, CC	*unknown*	AF139810			
*O. fusiforme*	CMW9968	Azerbaijan	*Populus nigra*		DQ294354		AY280461
*O. fusiforme*	CMW7131	Austria	*Quercus petraea*			AY280497	AY280464
*O. ips*	CBS 137.36	USA	*Ips* sp.				EU913724
*O. ips*	CMW7075	USA	*Ips integer*		DQ294381		DQ296101
*O. ips*	KUC2120		*Pinus* sp.	AY172021			
*O. karelicum*	CMW23099	Russia					EU443773
*O. lunatum*	CBS 112928, CMW10564	Austria	*Larix decidua*			AY280486	AY280467
*O. longisporum*	WIN(M)48			HQ634831			
*O. minus*	WIN(M)861			HQ634820			
*O. nigrocarpum*	CMW651	USA	*Pseudotsuga menziesii*		DQ294356		AY280480
*O. nigrocarpum*	Ci-203	Chile	*Pinus radiata*		EF506941		
*O. nigrocarpum*	CMW1468	Canada	*Dendroctonus ponderosae*			AF484457	
*O. novo-ulmi*	CMW10573	Austria	*Picea abies*		DQ294375		DQ296095
*O. phasma*	CMW20676	South Africa	*Protea laurifolia*				
*O. piceae*	AU100-1				AF234837		
*O. piceae*	CMW8093	Canada	*Tetropium* sp.				DQ296091
*O. piceae*	JCM6016			AB007663			
*O. pilliferum*	CMW7879	South Africa	*Pinus sylvestris*		DQ294377		
*O. pilliferum*	CBS 129.32	Germany		AJ243295			AF221628
*O. pilliferum*	CBS 158.74	Chile			DQ470955		
*O. quercus*	CMW3110	USA	*Junlans cinerea*		DQ294376		DQ296096
*O. quercus*	CMW 2467	France	*Quercus* sp.				
*O. quercus*	TUB F4272	Germany	*Xyleborus monographus*	AY497515			
*O. retusum*	ATCC22324	USA		HQ634841			
*O. ulmi*	CBS 298.87	Netherlands		M83261			
*O. ulmi*	CMW1462	USA	*Ulmus procera*		DQ294374		DQ296094
*A. macrospora*	CBS 367.53, C2231	Sweden	*Ips acuminatus*	EU170284			
*A. tingens*	CBS 366.53	Sweden	*Xyleborus glabratus* gallery	EU170277			
*O. gossypinum var. robustum*	MUCL18357	Spain	*-*			AY924388	
*O. stenoceras*	UCB 57.013		*-*	M85054			
*O. stenoceras*	CMW2530	Colombia	*Eucalyptus grandis*			AF484460	
*O. stenoceras*	CBS 139.51				DQ836904		
*O. stenoceras*	CBS 237.32 CMW3202	Norway	*Pinus* sp.				DQ296074
*O. stenoceras*	CBS 237.32	Norway	*Pinus* sp.				AY280471
*O. stenoceras*	CMW2524	South Africa	*Acacia mearnsii*			AF484459	
*O. stenoceras*		Germany	*Natrix natrix*				
*O. stenoceras*	CMW4007	Colombia	*Eucalyptus* sp. soil			AF484464	
*O. stenoceras*		Germany	*Python regius*				
*O.triangulosporum*	DSMZ4934	Brasil	*Araucaria araucaria*			AY934525	
*O.* *torulosum*	TUB F3258	Germany	*Trypodendron lineatum*	AY497517			
*S. lignivora*	CMW18600	South Africa	*Eucalyptus* sp.		EF139119		EF139104
*Ceratocystiopsis* sp.	AM434-K2G-1	USA	*Pinus* sp.			KT264634	
*Ceratocystiopsis* sp. *(Cop. minuta*-like)	Cop. sp. 1i	Brit. Columbia	*D. ponderosae* gallery		EU913667		EU913746
*Ceratocystiopsis* sp.	Cop. sp3i	Brit. Columbia	*Picea glauca* *,* *Ips perturbatus*		EU913676		EU913755
*Ceratocystiopsis* sp. *(Cop. manitobense*-like)	Cop. sp3ii	Brit. Columbia	*Ips perturbatus*			EU913717	EU913756
*Ceratocystiopsis* sp. *(Cop. minuta*-like)	YCC329	Japan	*L. kaempferi,* *Ips* sp.		EU913671		EU913750
*C. brevicomes*	CBS 333.97	USA	*Dendroctonus brevicomis*	HQ202311	EU913683		EU913761
*C. collifera*	WIN(M)908			HQ634832			
*C. collifera*	CBS 126.89	Mexico	*Dendroctonus valens*		EU913681	MH862160	EU913760
*C. concentrica*	WIN(M)53	Canada		HQ634849			
*C. concentrica*	WIN(M) 71-07	Canada			AF135571		
*C. manitobensis*	UM237	Canada	*Ips perturbatus*	EU984266	DQ268607	DQ268610	
*C. manitobensis*	UM237	Canada	*Ips perturbatus*			EU913714	
*C. manitobensis*	WIN(M)237	Canada		HQ634850			
*C. manitobensis*	CW13792	Canada	*Pinus resinosa*		DQ294358		DQ296078
*C. minuta*	RJ5095 (UM1533)	Poland	*Picea* sp. *Ips typographus*			EU913698	
*C. minuta*	RJ191(UM 1535)	Poland	*Picea* sp. *Ips typographus*			EU913700	
*C. minuta*	RJ705 (UM 1532)	Poland	*Picea abies* *Ips typographus*		EU913656		EU913736
*C. minuta*	CBS463.77	Mexico, USA	*Picea engelmanii*		EU913645		EU913725
*C. minuta*	WIN(M)1532	Canada		HQ634827			
*C. minuta*	YCC139	Japan	*Picea jezoensis* *Ips* sp.				EU913732
*C. minuta-bicolor*		South Africa	*Pinus* sp.		DQ294359		
*C. minuta-bicolor*	UAMH9551	Canada	*Pinus contorta*				
*C. minuta-bicolor*	WIN(M)480	Canada		HQ634848			
*C. minuta-bicolor*	CBS635.66 (UM844)	USA	*Pinus contorta*				EU913745
*C. minuta-bicolor*	UM 480	Canada	*Pinus contorta*			EU913705	
*C. minima*	WIN(M)85	Canada		HQ634856			
*C. minima*	CBS 182.86	USA	*Pinus banksiana*		EU913663	EU913706	EU913743
*C. pallidobrunnea*	UM51	Canada	*Populus tremuloides*		EU913682		
*C. pallidobrunnea*	WIN(M)51	Canada		HQ634842			
*C. parva*	JR71-21	Canada		HQ595735			
*C. ranaculosa*	CMW13940	USA	*Pinus echinata*		DQ294357		DQ296077
*C. ranaculosa*	WIN(M)919			HQ634840			
*C. ranaculosa*	CBS 216.88	USA					
*C. rollhanseniana*	CW13791	Norway	*Pinus sylvestris*		DQ294362		DQ296082
*C. rollhanseniana*	WIN(M)110	Canada		HQ634834		EU913719	
*C. rollhanseniana*	UM110	Norway	*Pinus sylvestris*				EU913758

^a^ PC—fungal strains obtained from *Platypus cylindrus* and its galleries on *Quercus suber*, MEAN culture collection of INIAV Institute, Oeiras, Portugal; C, Iowa State University, Department. of Plant Pathology, USA; CBS, Culture collection of the Westerdijk Fungal Biodiversity Institute, the Netherlands; ^b^ Accession numbers of sequences newly produced (bold). SSU rDNA: small subunit region of the ribosomal RNA gene; LSU rDNA: large subunit region of the ribosomal RNA gene, 5.8S-ITS2: internal transcribed spacer 2 and TUB: β-Tubulin.2.3. PCR, Sequencing and Phylogenetic Analyses.

## Data Availability

The generated sequences are in GenBank^®^ nucleic acid sequence database, the National Center for Biotechnology Information (NCBI) at https://www.ncbi.nlm.nih.gov/, accessed on 6 April 2022.

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
