# Peer review of "Ceratocystiopsis quercina sp. nov. Associated with Platypus cylindrus on Declining Quercus suber in Portugal"

_biology, 2022, doi:10.3390/biology11050750_

Round 1
Reviewer 1 Report
This manuscript describe a new species of Ceratocystiopsis (Ceratocystiopisis quercina) associated with Platypus cylindrus on Quercus suber in Portugal. The manuscript is interesting and present information how species discovered and identified, how described by morphology and verified on molecular side using four loci phylogeny. The paper is well written and seem to be well supported by morphology (pictures)/description and Phylogeny trees. My only concern is since this was found in declining Quercus, so not much information about pathogenicity, no koch postulates tried or attempt here to see if the pathogen was causing plant host disease (or reinfection), just some info about some discoloration, why not done? This could have bring more support of the finding. Also other closely related organism in quercus or in blast match with some level of mismatch with some region as Raffalela quercina or Dryadomyces montetyi with the ITS region, why none of them been add to the phylogeny trees? Your clade is very differentiated from other Ceratocystiopsis species from the trees?
Specific comments:
P12 Species description, not sure in Biology, but usually new species description in latin is need for species description in many journals?
Figure 3, j and k very week difficult to see?
Author Response
Dear Reviewer,
Thank you for the careful and thorough reading of our manuscript “Ceratocystiopsis quercina sp. nov. associated with Platypus cylindrus (Coleoptera: Curculionidae) on declining Quercus suber in Portugal”.
We appreciate the thoughtful comments and constructive suggestions, which help us to improve the quality of this manuscript. We have tried to reply to each of the comments and we are now convinced that our MS meets the requirements to be published in this Journal.
As we were asked, all revised items are marked along the text. Also, the answers to reviewers’ comments are written below.
Thank you once again for your advice and review.
The authors
Response to reviewer #1 comments (please see the attachement)
My only concern is since this was found in declining Quercus, so not much information about pathogenicity, no Koch postulates tried or attempt here to see if the pathogen was causing plant host disease (or reinfection), just some info about some discoloration, why not done? This could have brought more support of the finding.
Dear Reviewer, we understand your concerns and you are right: pathogenicity tests confirming the virulence of Cop. quercina against cork oak would bring more importance to our finding. And in true we performed these tests, by the time we did it for other species (in Inácio et al. 2012 OILB bulletin; data not published). All these data were never published in IF journals because of their limited information, as we have been told. The fact that cork oak is our national tree and it is completely impossible to inoculate adult trees, only allowed us to conduct these tests in seedlings which is not informative about the complex pathosystem involving ophiostomatoid fungi and their vector Platypus cylindrus, we understand that. Thus, we chose not to include these limited results, similarly to what the authors of parallel works did. We know for sure, after years evaluating declining cork oaks, that Cop. quercina is always present in the beetles’ galleries, causing extensive discoloration of the wood. Furthermore, we confirmed that Cop. quercina isolates were also obtained from declining cork oaks with visible aborted attacks of P. cylindrus (Inacio et al. 2011). So, even if insects did not succeed to colonize the tree they are able to inoculate the pathogen into a susceptible host, which in turn is able to cause tree wilting alone, as other Ophiostoma s. l. do.
Also other closely related organism in quercus or in blast match with some level of mismatch with some region as Raffalela quercina or Dryadomyces montetyi with the ITS region, why none of them been add to the phylogeny trees? Your clade is very differentiated from other Ceratocystiopsis species from the trees?
Dear reviewer, thank you for this pertinent question. The main goal of our research was to include sequences of Ophiostoma and Ceratocystiopsis species for each of the several analysed DNA regions to confirm the placement of our new species in the Cop. clade and not in the closely related Ophiostoma clade. The species R. quercina and D. montetyi are in fact ophiostomatoid included in the same pathosystem cork oak-beetle-ambrosia fungi but morphologically and molecularly distinct from this new species, therefore were not included in the trees. The analysis conducted using several DNA regions allowed for the discrimination of Cop. quercina, clearly differentiated for other existent Cop. species.
Species description, not sure in Biology, but usually new species description in Latin is need for species description in many journals?
Dear Reviewer, you are right, un Latin description was needed for new fungal species but not anymore. Please see Aime et al 2021 “How to publish a new fungal species, or name, version 3.0”: Latin is no longer required for descriptions and/or diagnoses, and these can now be either in English or Latin but not in any other language (Art. 39.2).
Figure 3, j and k very week difficult to see
You are right and we are so sorry about that but we do not have better pictures, and we will never have since our optical equipment is not of high performance.
Once again, we thank you for the questions and suggestions.
Best regards,
Reviewer 2 Report
Dear authors thank you for your collaborations to the understanding of new species in pathosystems. In general, I found the manuscript well written with only a few areas that need clarification, please read carefully each sentence to confirm the correct message its being transmitted. I made several comments but there are sure other stances that may need attention.
All the best
Specific comments
- There are a few grammatical errors that need to be address throughout.
- Line 58-69 Consider joining the last 3 paragraphs in the introduction, by moving the middle one to the top.
- Check tense in line 62
- Line 77. Were there 12 trees within the two plots or 12 at each plot? please clarify
- Line 83. What magnification you used? what is an iris scissor, who makes it? Overall, more detail is required from your collection procedure (paragraph 1) for someone trying to replicate your study.
- Line 71 change “their” to “its” as you talk about the species
- Line 173 check italicization here and throughout
- Line 175. You mention Rafaella, but you are not explaining why?, please clarify?
- Line 219. Consider calling this “Collection” or “known distribution”
- Line 259. Check size format throughout
- Line 268. Consider developing this section into a diagnostic section.
- Line 275. …X number of species were found including a new spp. Describing a new spp. was not your initial objective.
- Line 289 consider the use of “important” and changing “food” to “alimentary”
- Line 315, I am sure this is not a new strategy and have read about it before.
Author Response
Dear Reviewer,
Thank you for the careful and thorough reading of our manuscript “Ceratocystiopsis quercina sp. nov. associated with Platypus cylindrus (Coleoptera: Curculionidae) on declining Quercus suber in Portugal”.
We appreciate the thoughtful comments and constructive suggestions, which help us to improve the quality of this manuscript. We have tried to reply to each of the comments and we are now convinced that our MS meets the requirements to be published in this Journal.
As we were asked, all revised items are marked along the text. Also, the answers to reviewers’ comments are written below.
Thank you once again for your advice and review.
The authors
Response to reviewer #2 comments
- There are a few grammatical errors that need to be address throughout.
Dear Editor, thank you for pointing us these mistakes! We read carefully the manuscript and tried to correct them.
- Line 58-69 Consider joining the last 3 paragraphs in the introduction, by moving the middle one to the top.
You are right and the text makes more sense now. Because of this modification, the respective references had to be modified and we did it (yellow highlight in the references list).
- Check tense in line 62
We checked and modified it as per your suggestion.
- Line 77. Were there 12 trees within the two plots or 12 at each plot? please clarify
It has been clarified as you suggested, thank you. “Only” 12 trees were cut in total because cork oak is our national tree and it is rather difficult to have them cut! No authorization is given even if they are declining and sampling is always limited.
- Line 83. What magnification you used? what is an iris scissor, who makes it? Overall, more detail is required from your collection procedure (paragraph 1) for someone trying to replicate your study.
Dear reviewer, you are absolutely right, thank you for your questions! We added the missing information and better clarified the material we used.
- Line 71 change “their” to “its” as you talk about the species
It has been changed as per your suggestion.
- Line 173 check italicization here and throughout
We used the MDPI template and the respective format.
- Line 175. You mention Rafaellea, but you are not explaining why? please clarify?
You’re right and we add the sentence which now makes much more sense, thank you.
- Line 219. Consider calling this “Collection” or “known distribution”
Thank you, we cut unnecessary information.
- Line 259. Check size format throughout
We used the MDPI template and the respective format.
- Line 268. Consider developing this section into a diagnostic section.
We used the format other authors use for new fungal species descriptions at MDPI journals.
- Line 275. …X number of species were found including a new spp. Describing a new spp. was not your initial objective.
Dear reviewer, of course you are absolutely right: finding a new species was not our primary goal. Instead, we aimed at clarifying the factors involved in cork oak decline but a new fungal species was found which in our opinion deserves to be spread.
- Line 289 consider the use of “important” and changing “food”cto “alimentary”
We modified as per your suggestion
14. Line 315, I am sure this is not a new strategy and have read about it before.
Thank you for pointing us this issue: we have been more cautious and added “probably”.
Once again, dear Reviewer, thank you for your detailed review which helped us improving our manuscript. Best regards,
Reviewer 3 Report
The paper is interesting, well written and contains quite original data which look properly analyzed. Just one note related to systematics: all the species (insects, plants, fungi....) when reported for the first time in the text, should be reported with Authority and systematics.
Author Response
Dear Reviewer,
Thank you for the careful and thorough reading of our manuscript “Ceratocystiopsis quercina sp. nov. associated with Platypus cylindrus (Coleoptera: Curculionidae) on declining Quercus suber in Portugal”.
We appreciate the thoughtful comments and constructive suggestions, which help us to improve the quality of this manuscript. We have tried to reply to each of the comments and we are now convinced that our MS meets the requirements to be published in this Journal.
As we were asked, all revised items are marked along the text. We followed your suggestion and added the authorities for the species when first mentioned, except for the tables.
Thank you once again for your advice and review.
The authors